

# *Trichoplax adhaerens* reveals a network of nuclear receptors sensitive to 9-*cis*-retinoic acid at the base of metazoan evolution

Jan Philipp Novotný[1,2], Ahmed Ali Chughtai[1], Markéta Kostrouchová[1,3], Veronika Kostrouchová[1], David Kostrouch[1], Filip Kaššák[1], Radek Kaňa[4], Bernd Schierwater[5,6], Marta Kostrouchová[1] and Zdenek Kostrouch[1]

[1] Biocev, First Faculty of Medicine, Charles University, Vestec, Czech Republic
[2] Department of Medicine V., University of Heidelberg, Heidelberg, Germany
[3] Department of Pathology, Third Faculty of Medicine, Charles University, Prague, Czech Republic
[4] Institute of Microbiology, Laboratory of Photosynthesis, Czech Academy of Sciences, Třeboň, Czech Republic
[5] Institute for Animal Ecology and Cell Biology, University of Veterinary Medicine, Hannover, Germany
[6] Department of Ecology and Evolutionary Biology, Yale University, New Haven, CT, United States of America

Corresponding author
Zdenek Kostrouch,
Zdenek.kostrouch@lf1.cuni.cz

## ABSTRACT

*Trichoplax adhaerens*, the only known species of Placozoa is likely to be closely related to an early metazoan that preceded branching of Cnidaria and Bilateria. This animal species is surprisingly well adapted to free life in the World Ocean inhabiting tidal costal zones of oceans and seas with warm to moderate temperatures and shallow waters. The genome of *T. adhaerens* (sp. Grell) includes four nuclear receptors, namely orthologue of RXR (NR2B), HNF4 (NR2A), COUP-TF (NR2F) and ERR (NR3B) that show a high degree of similarity with human orthologues. In the case of RXR, the sequence identity to human RXR alpha reaches 81% in the DNA binding domain and 70% in the ligand binding domain. We show that *T. adhaerens* RXR (TaRXR) binds 9-*cis* retinoic acid (9-*cis*-RA) with high affinity, as well as high specificity and that exposure of *T. adhaerens* to 9-*cis*-RA regulates the expression of the putative *T. adhaerens* orthologue of vertebrate L-malate-NADP$^+$ oxidoreductase (EC 1.1.1.40) which in vertebrates is regulated by a heterodimer of RXR and thyroid hormone receptor. Treatment by 9-*cis*-RA alters the relative expression profile of *T. adhaerens* nuclear receptors, suggesting the existence of natural ligands. Keeping with this, algal food composition has a profound effect on *T. adhaerens* growth and appearance. We show that nanomolar concentrations of 9-*cis*-RA interfere with *T. adhaerens* growth response to specific algal food and causes growth arrest. Our results uncover an endocrine-like network of nuclear receptors sensitive to 9-*cis*-RA in *T. adhaerens* and support the existence of a ligand-sensitive network of nuclear receptors at the base of metazoan evolution.

# INTRODUCTION

Life on Earth began 4.1 to 3.5 billion years ago (*Bell et al., 2015*) with the appearance of the first unicellular organisms that subsequently evolved, in part, to multicellular lifeforms forming the kingdom Metazoa that have specialized tissues for digestion, regulation of homeostasis, locomotion, perception, analysis of the environment and reproduction.

In contrast to unicellular organisms, metazoans are in need of regulatory mechanisms that provide the means of coordination between various tissues in a tight arrangement with cellular homeostasis. This coordination on the level of humoral signaling includes regulation by nuclear receptors (NRs), which respond to small, mostly hydrophobic molecules, including hormones produced by specific tissues, metabolites or even molecules present in the environment and transfer these signals to the nucleus, thus leading to a dynamically changing but adaptive gene expression (*Escriva, Bertrand & Laudet, 2004*).

NRs therefore play an important role in maintaining intra- and inter-cellular functions in multicellular organisms. Their overall structure is common in most nuclear receptors and consists of an A/B (N-terminal) domain, the DNA binding domain (DBD), a hinge region, the ligand binding domain (LBD) and the C-terminal domain (*Kumar & Thompson, 1999*; *Robinson-Rechavi, Escriva Garcia & Laudet, 2003*). The DBD and LBD of NRs exhibit an especially high degree of conservation and the changes that were acquired during evolution allow classification of the NR protein family into six subfamilies (*Laudet, 1997*; *Escriva et al., 1998*). NRs that evolved within these subfamilies show functional connections that include specialization of regulatory functions in time or cell type restriction, fortification of ancestral functions or their specific inhibition by newly evolved NRs (*Escriva, Bertrand & Laudet, 2004*; *Kostrouchova & Kostrouch, 2015*).

With the overall structure maintained across metazoan species, nuclear receptors show significant heterogeneity regarding their quantity and function, many of which have not yet been explored in e.g., *Caenorhabditis elegans* with over 280 nuclear receptors (reviewed in *Kostrouchova & Kostrouch, 2015*).

The evolutionary changes accumulated in diversified NRs allow functional subspecialization at the level of specific sequence binding within gene promoters (response elements), protein-protein interactions with functionally linked receptor interactors and adoption of new ligands as specific hormonal regulators. The evolution of hormonal ligands acquired by different species during evolution is well documented and indicates the potential of NRs to adopt new ligands as regulators (*Escriva, Delaunay & Laudet, 2000*; *Markov & Laudet, 2011*). Indeed, it is now thought that NRs evolved as environmental sensors that were able to sense a wide variety of compounds with low affinity and specificity, some of which later-on acquired higher affinity binding towards certain ligands that are products of metabolic pathways (*Holzer, Markov & Laudet, 2017*). This can be exemplified by the high affinity and specificity binding of certain receptors, such as the mineralocorticoid or androgen receptors, while the family of PPARs shows a rather promiscuous binding to a variety of different substances (*Issemann & Green, 1990*). Keeping in mind the metabolic origin of NR ligands, it is not completely surprising to see different ligands binding to

orthologues across species, such as Triac and T3 in the case of TR (*Paris et al., 2008*), thus changing in the course of evolution and adapting to new environments.

This is accompanied by the essential questions, to what degree the plasticity of ligand selection is a fundamental property of NRs and what the origin of specific ligand binding by NRs might be. It has been suggested that the original NR, which is the ancestral NR possessing gene regulatory capacity, may have been an unliganded molecular regulator (*Escriva et al., 1997*). However, it is now believed that the ancestral NR is most closely related to the NR2 subfamily, as members of this family can be found in basal metazoans and are sensors of fatty acids (*Holzer, Markov & Laudet, 2017*). More recently, it was proposed that the ligand binding and ligand-dependent regulatory potential of NRs is an inherent feature of the evolution of NRs (*Bridgham et al., 2010*). Due to their nature of fine tuning cellular responses in response to environmental changes without necessarily showing high affinity binding to a set of ligands, cross-species comparison of nuclear receptor networks might shed light on the details of the NR network function (*Holzer, Markov & Laudet, 2017*).

A search for NRs that may be closely related to an ancient ancestor of the NR family led to the discovery of an RXR orthologue in Cnidaria (*Kostrouch et al., 1998*). Surprisingly, this receptor showed not only extremely high degree of sequence identity with vertebrate RXRs, far surpassing the degree of conservation observed in insects but also by its ability to bind the same ligand as vertebrate RXRs, *9-cis*-retinoic acid (*9-cis*-RA), with an affinity close to that reported for vertebrate RXRs. Similarly, as vertebrate RXRs, the jellyfish RXR showed a specific binding preference for *9-cis*-RA over all-*trans*-retinoic acid (AT-RA) and was able to heterodimerize with vertebrate thyroid hormone receptor alpha. Recent genome sequencing projects confirmed the existence of highly conserved RXR across several metazoan species including insects (*Locusta migratoria* (*Nowickyj et al., 2008*)), that are evolutionarily older than species with a more diversified RXR orthologue such as Usp found in *Drosophila* (reviewed in *Gutierrez-Mazariegos, Schubert & Laudet, 2014*).

To date, the nuclear receptor network has mainly been studied in complex organisms already in possession of an extensive endocrine network. Albeit dissection of nuclear receptor networks in these organisms can outline functions and associated regulatory cascades, basal tasks might be obscured by the gain of further, more specialized functions. Genome analysis of the basal metazoan *Trichoplax adhaerens* by whole genome sequencing revealed four highly conserved nuclear receptors, namely orthologues of HNF4 (NR2A), RXR (NR2B), ERR (NR3B) and COUP-TF (NR2F) (*Baker, 2008*; *Srivastava et al., 2008*) and thus allows assessment of the most basal workings of nuclear receptors. Surprisingly, the degree of conservation of the predicted placozoan NRs with known vertebrate NRs is not only very high at the level of the predicted secondary structure, as can be expected for true NRs, but also at the level of the primary amino acid sequence. Especially the similarity of the placozoan RXR (TaRXR) to its vertebrate orthologues is high, as it is in the case of the cubomedusan RXR. *T. adhaerens,* which shows characteristics of a basal metazoan with only a few cell types (*Smith et al., 2014*) and a relatively simple 4 member NR complement, offers a unique model that may shed light on the evolution of gene regulation by NRs.

In this presented work, we attempted to study the placozoan RXR orthologue functionally. Our results show that *T. adhaerens* RXR binds 9-*cis*-RA with an affinity comparable to that of vertebrate and jellyfish RXRs and that *T. adhaerens* responds to nanomolar concentrations of 9-*cis*-RA with a transcriptional upregulation of the putative orthologue of a malic enzyme that is regulated by a heterodimer formed by liganded thyroid hormone receptor and RXR in vertebrates. We also show that 9-*cis*- RA affects the relative expression of the four NRs present in *T. adhaerens* genome suggesting that these NRs may form a regulatory network capable of responding to possible ligands present in these animals or their environment. In line with this, growth, multiplication and appearance of *T. adhaerens* are strongly affected by food composition, especially by red pigment containing algae suggesting that specific food components or their metabolites may be ligands involved in the ancestral regulatory network of NRs. In support of this, 3.3 nM 9-*cis*-RA interferes with *T. adhaerens* growth response to the feeding by *Porphyridium cruentum* and causes balloon-like phenotypes and death of animals while animals fed by *Chlorella sp.* are partially protected against the treatment by 3.3 nM 9-*cis*-RA, do not develop balloon-like phenotypes but are also arrested in their growth and propagation indicating that 9-*cis*-RA interferes with *T. adhaerens* growth and development.

## METHODS

### Bioinformatics and cloning of RXR

The predicted RXR gene models on jgi (http://jgi.doe.gov/) (*Nordberg et al., 2014*) were screened for the characteristic molecular signature of the DNA binding domain (C-X2-C-Xl3-C-X2-C-X15-C-X5-C-X9-C-X2-C-X4-C-X4-M) (*Kostrouch, Kostrouchova & Rall, 1995*) and the appropriate predicted gene model (protein ID 53515) was selected for further use.

The alignment of different RXRs was performed by Clustal Omega (http://www.ebi.ac.uk/Tools/msa/clustalo/) (*Sievers et al., 2011*) and adjusted/exported as an image file using Jalview (http://www.jalview.org). Protein domain characterization was performed with SMART (*Schultz et al., 1998*; *Letunic, Doerks & Bork, 2015*). Analysis of HNF4, ERR and COUP-TF was done similarly. Phylogenetic analysis was performed on RXR ClustalO alignment using PhyMLv3.1 (*Guindon et al., 2010*) implemented in SeaView v4.6.1 with a 100 bootstrap analysis and SPR distance computation. The tree was then visualized using FigTree v1.4.3.

*T. adhaerens* total RNA was obtained from 50-100 pooled individual animals and extracted using TRIZOL® reagent (Invitrogen, Carlsbad, CA, USA) according to the protocol supplied by the manufacturer.

Subsequently, cDNA was prepared with random hexamers and SuperScript III (Invitrogen™) according to the manufacturer's protocol. Several RXR transcripts were then amplified by PCR with primers covering the starting sequence ((GCG-GATCC)ATGGAGGACAGATCGTTTAAAAAA), starting at 32 bp 5′ of ATG (TCTACCAATGTTTATCGCATCGGTTA) and starting at 97 bp 5′ of ATG (TTAAGGCT-TAACTGATGATGTTGTGAATG) with a common reverse primer covering the last 24 bp of the predicted gene sequence ((CGGAATTC)TTAAGAACTGCCTGTTTCCAGCAT).

Each PCR product was then ligated into pCR®2.1-TOPO® or pCR®4-TOPO® vector with the classic TA Cloning Kit and TOPO TA Cloning Kit (Invitrogen™), respectively. The ligated products were then transformed using One Shot® TOP10 Chemically Competent *E. coli* and cultured on LB Agar plates containing 100 µg/ml ampicillin. Plasmid DNA was extracted from obtained colonies and screened for mutations by sequencing using vector specific M13 forward and reverse primers. Only non-mutated sequences were used in subsequent experiments. The RXR fragments were then restricted and inserted into pGEX-2T vector system for bacterial expression (Addgene, Cambridge, MA, USA). Proper insertion was verified by sequencing.

## Protein expression

Bl21 pLysS bacteria were transformed with previously described RXR mRNA inserted into pGEX-2T vector. Stocks of transformed bacteria were stored in 8% glycerol according to the Novagen pET System Manual (11th edition) (https://www.google.cz/search?q=Novagen+pET+System+Manual+&ie=utf-8&oe=utf-8&client=firefox-b&gfe_rd=cr&ei=T9z1WMHJDsni8AfpmoGoCQ). For protein expression, bacteria were scraped from stock and incubated in Liquid Broth (LB) with ampicillin (100 µg/ml) and chloramphenicol (34 µg/ml) overnight. The culture was then used to inoculate 100 ml of LB + antibiotics and grown to OD600 = 0.6–0.8 at 37 °C, then induced with 100 µl 1M IPTG (isopropyl-D-thiogalactopyranoside) (Sigma-Aldrich, St. Louis, MO, USA) and moved to 25 °C (RT) for 5 h. The culture was then spun at 9,000 xg for 15 min and the supernatant discarded. The bacterial pellet was resuspended in 10 ml GST binding buffer (25 mM Tris pH 7.5, 150 mM NaCl, 1 mM EDTA) + protease inhibitor (S8820 Sigma Fast, Sigma-Aldrich, St. Louis, MO, USA or cOmplete™, EDTA-free Protease Inhibitor Cocktail, Roche, Basel, Switzerland). Bacteria were then lysed by 6 × 20 s ultrasonication on ice (50 watts, 30 kHz, highest setting—100%) (Ultrasonic Processor UP50H, Hielscher Ultrasonics GmbH, Teltow, Germany) and subsequently incubated with 15–20 mg glutathione agarose beads (Sigma-Aldrich®) prepared according to manufacturer's instructions. Incubation took place at 4 °C for about 10 h after which the beads were washed according to instructions, resuspended in regeneration buffer (50mM Tris–HCl pH7.4, 1mM EDTA, 120 mM KCl, 5 mM DTT, 8% glycerol (v/v)) or 50mM TRIS–HCl pH 7.4 + 9% (v/v) glycerol for subsequent thrombin (bovine plasma, Sigma-Aldrich®) cleavage, if performed, and then adjusted for regeneration buffer conditions. GST-TaRXR was eluted from glutathione agarose beads using 10 mM reduced glutathione (Sigma-Aldrich, StLouis, Mo, USA) in 50 mM Tris–HCl buffer pH 8.0. The size of the GST-TaRXR fusion protein was checked by polyacrylamide gel electrophoresis. Thrombin cleavage was performed at RT for 4 h and the quality of the purified protein was assessed by polyacrylamide gel electrophoresis.

## Radioactive *9-cis RA* binding assay

Radioactive $^3$H-labelled 9-*cis-RA* and $^3$H-labelled AT-RA were purchased from PerkinElmer (Waltham, MA, USA). Binding was performed in 100 µl binding buffer (50mM Tris–HCl pH7.4, 1mM EDTA, 120 mM KCl, 5 mM DTT, 8% glycerol (v/v), 0.3%

to 0.5% (w/v) CHAPS (3-[(3-Cholamidopropyl)dimethylammonio]-1-propanesulfonate hydrate, Sigma-Aldrich)) for 2 h on wet ice in a dark environment. The protein used for binding was either GST-RXR fusion protein on beads with about 375 ng/assay and thrombin-cleaved RXR. For estimation of specific binding, 200x excess of either 9-*cis*- RA or AT-RA (Sigma-Aldrich) was used. In case of GST-RXR fusion protein, 50 µl of the supernatant was removed after 30 s at 1300 g and washed 3x with 1000 µl wash buffer (50 mM Tris–HCl pH7.4, 1 mM EDTA, 120 mM KCl, 5 mM DTT, 8% (v/v) glycerol, 0.5% (w/v) CHAPS) removing 900 µl after each wash. For cleaved RXR protein 10 µl hydroxyapatite slurry (AG-1 XB Resin, Bio-Rad, Hercules, CA, USA) suspended in binding buffer (12.7 mg/100 µl) were added to the assay and mixed twice, collecting the apatite slurry by centrifugation (15 s at 600 g). 95 µl of the supernatant was removed and the slurry washed twice with 1 ml of wash buffer, removing 900 µl after each wash. Work with retinoids was done under indirect illumination with a 60 W, 120 V yellow light bulb (BugLite, General Electric Co, Nela Parc, Cleveland Oh, USA) as described (*Cahnmann, 1995*). The radioactivity of the GST-fusion protein and cleaved protein was measured using Packard Tri-Carb 1600TR Liquid Scintillation Analyzer (Packard, A Canberra Company, Canberra Industries, Meriden, CT, USA) and Ultima Gold Scintillation Fluid (PerkinElmer, Waltham, MA, USA). The fraction of bound $^3$H-labelled 9-cis-RA and $^3$H-labelled all-*trans*-RA was determined as a ratio of the bound radioactivity of precipitated GST-TaRXR/total radioactivity used at the particular condition (determined as the sum of bound radioactivity and the total radioactivity of collected wash fluids) in the absence of non-radioactive competitors or 200 fold excess of 9-cis-RA and all-trans-RA in the case of $^3$H-labelled 9-*cis*-RA and 40 fold excess of non-radioactive competitors in the case of $^3$H-labelled all-*trans*-RA (to compensate for the higher affinity of 9-*cis*-RA compared to all-*trans*-RA in binding to TaRXR ).

## Culture of *T. adhaerens* and algae

*Trichoplax adhaerens* was cultured in Petri dishes containing filtered artificial seawater (Instant Ocean, Spectrum Brands, Blacksburg, VA, USA) with a salinity of approx. 38–40 ppt. *Rhodomonas salina* (strain CCAP 978/27), *Chlorella sp*., *Porphyridium cruentum* (UTEX B637) and other non-classified algae, as well as aquarium milieu established in the laboratory by mixing salt water obtained from a local aquarium shop were used to maintain the stock. The cultures were kept at approx. 23 °C and an automated illumination for 12 h/day was used with a conventional light bulb on a daylight background from late spring to mid-summer in the laboratory located at 50.07031N, 14.42934E with laboratory windows oriented eastward. The natural illumination included almost direct morning light from 8 AM to 10.30 AM, indirect sunlight for most of the daytime and sunlight reflected from a building across the street from 1 PM to 6 PM. Algae were maintained as described (*Kana et al., 2012*; *Kana et al., 2014*). The experiments were performed predominantly during sunny weather.

## Treatment of *T. adhaerens* with retinoic acids

Incubation of the animals was done overnight in the absence of light. Each batch within an experiment was derived from similar cultures and fed with similar amounts and

 

composition of algae. All experiments were started in a dark room with indirect yellow light illumination (similarly as in the case of the ligand binding studies) and further incubations were done in the dark for 24 h. In experiments aimed at the visualization of 9-*cis*-RA effect on *T. adhaerens* response to feeding conditions, parallel cultures were set and fed with *P. cruentum*. Large animals of approximately the same size were individually transferred to new control and experimental cultures and fed with *P. cruentum* algal cells. After 6 h of incubation under natural indirect illumination, all animals in both control and experimental cultures were photographed (max. magnification on Olympus SZX7 with Olympus E-410 camera) and the final volume of cultures was adjusted to 50 ml (determined by the weight of cultures in 110 mm glass Petri dishes). Next, the room was darkened and further manipulations were done under indirect illumination with yellow light. Five µl of vehicle (1% DMSO in ethanol) or vehicle containing 9-*cis*-RA was added into 50 ml of final volume to the final concentration of 9-*cis*-RA 3.3 nM. Similarly, parallel sub-cultures were prepared from slowly growing cultures fed by microorganisms covering glass slides in an equilibrated 25 l laboratory aquarium and fed by *Chlorella sp*. Cultures were incubated in the dark for 24 h and all animals were counted under microscope and photographed again. The cultures were then left under natural illumination and cultured for an additional two or three weeks. Animals fed by *P. cruentum* were measured again at 72, 90 and 450 h and those fed by *Chlorella sp*. at 72, 90 and 378 h.

## Quantitative PCR

Droplet digital PCR was performed on a QX100 Droplet Digital PCR System (Bio-Rad Laboratories, Hercules, CA, USA). For this, *T. adhaerens* was cultured according to culture conditions described and 4-10 animals removed per 100 µl TRIZol reagent (Invitrogen, Carlsbad, CA, USA). Total RNA was measured by a UV spectrophotometer and used as a reference for normalization.

Reverse transcription was performed with SuperScript III Reverse Transcriptase (ThermoFisher, Waltham, MA, USA) according to manufacturer's instructions. The cDNA (corresponding to 100–500 ng of RNA) was then mixed with ddPCR Supermix (Bio-Rad, Hercules, CA, USA) according to the manufacturer's instructions and analyzed. PCR primers were designed using the UPL online ProbeFinder (Roche) software and were as follows:

TaRXR—left:tctgcaagttggtatgaagca, right: agttggtgtgctattctttacgc
TaHNF4 ref|XM_002115774.1|:
left: ggaatgatttgattttacctcgac, right: tacgacaagcgatacgagca
  TaCOUP-TF (ref|XM_002109770.1|):
left: attttgaatgctgcccaatg, right: ttactggttgtggagtatggaaac
TaSoxB1 (ref|XM_002111308.1|):
left: tgtcagatgcggataaacga, right: ggatgttccttcatgtgtaatgc
TaTrox-2 (ref|XM_002118165.1|):
left: gcctatagtcgatcctgccata, right: ttggtgatgatggttgtcca
TaPaxB1 (gb|DQ022561.1|):
left: tcaaacgggttctgttagcc, right: ggtgttgccaccttaggc

TaERR (nuclear receptor 3, gb|KC261632.1):
left: ttacgcatgtgatatggttatgg, right: agcgtgcctatttatttcgtct

Results were subsequently analyzed using the Bio-Rad ddPCR software. Manual correction of the cut off was performed when automated analysis was not possible. To visualize changes in nuclear receptor expression in the absence of a reliable housekeeping gene as a reference, we considered the absolute quantity of each nuclear receptor as a percentage of the overall nuclear receptor expression and subsequently visualized the change of receptor expression by subtraction of the percentage of the control experiment. Absolute copy numbers of the proposed malic enzyme orthologue in *T. adhaerens* have been normalized to overall RNA quantity for expressional analysis.

Experiments with quantification by qRT-PCR were performed on a Roche LightCycler II with OneTaq polymerase and the same probes as for ddPCR.

For the estimation of the relative expression of NRs in small (<0.5 mm) versus big animals (>1 mm), 20 to 30 animals from the same culture were used for each paired experiment.

## Identification of *T. adhaerens* orthologue of L-malate-NADP$^+$ oxidoreductase (EC 1.1.1.40)

P48163 (MAOX_HUMAN) protein sequence was used as the query sequence and searched against *T. adhaerens* database with BLASTP on http://blast.ncbi.nlm.nih.gov/Blast.cgi using standard algorithm parameters. The best hit was a hypothetical protein TRIADDRAFT_50795 with a sequence identity of 57% and a query coverage of 93% and was assumed to be *T. adhaerens* closest orthologue of vertebrate L-malate-NADP$^+$ oxidoreductase.

## Microscopy and image analysis

Observation of *T. adhaerens* was done with an Olympus SZX10 microscope equipped with DF Plan 2x objective and Olympus DP 73 camera operated by CellSens Dimension computer program (kindly provided by Olympus, Prague, Czech Republic) or Olympus CKX41 or SZX7 with Olympus E-410 camera and QuickPhoto Micro 3.1 program.

Circularity was calculated by establishing the area ($A$) and perimeter ($p$) of *T. adhaerens* using ImageJ (https://imagej.nih.gov/ij/) and then calculated with the isoperimetric quotient $Q = \frac{4\pi A}{p^2}$, ($A$, Area; $p$, perimeter). GraphPad Prism 5 (or higher) was used for graphical representation and calculations of the confidence intervals with $p = 0.05$.

## RESULTS

### *T. adhaerens* retinoid X receptor shows high cross-species sequence identity

By using the ab initio model of the JGI *Trichoplax* database as a reference, we screened the *Trichoplax* JGI database for RXR orthologues with a complete DBD and LBD sequence and were able to obtain, as well as verify a full length RXR transcript previously not annotated as the 'best model'. Blastp analysis showed a high sequence similarity to human, as well as mouse RXR with 66% overall sequence identity to human RXR alpha.
SMART analysis of the proposed TaRXR sequence showed a zinc finger DNA binding domain (amino acid residues 16–87) and a ligand binding domain (amino acid residues 155–342) with E values $<10^{-40}$. Blast analysis of the zinc finger DNA binding and ligand binding domains revealed a sequence identity of 81% and 70% to human RXR alpha, respectively. Both domains contained the predicted molecular pattern characteristic for each domain. The heptad repeat LLLRLPAL proposed for dimerization activity (*Forman & Samuels, 1990b*; *Forman & Samuels, 1990a*; *Kiefer, 2006*) as well as the LBD signature for 9-*cis*-RA binding Q-x(33)-L-x(3)-F-x(2)-R-x(9)-L-x(44)-R-x(63)-H (*Egea, Klaholz & Moras, 2000*) were present (Fig. 1). From 11 amino acid residues shown to be critical for 9-*cis*-RA binding (A271, A272, Q 275, L 309, F 313, R 316, L 326, A 327, R 371, C 432, H 435), nine are conserved, while the remaining two amino acids are substituted (A327S, and C432T (C432A in *Tripedalia cystophora*)). Due to the high sequence identity, we propose a 9-*cis*-retinoic acid binding capability of the hypothesized TaRXR sequence, as well as DNA binding capability. Phylogenetic analysis using PhyML algorithm indicates that TaRXR is likely to precede branching of RXRs in cubomedusae and scyphomedusae and clusters with RXRs in bilateria Fig. S1).

The remaining three NRs identified in the *T. adhaerens* genome show also very high overall sequence identity with vertebrate orthologues. Alignments of *T. adhaerens* HNF4, COUP-TF and ERR with orthologues from selected species can be found in File S2.

## TaRXR shows preferential binding affinity to *9-cis* retinoic acid over all-*trans*-retinoic acid

In order to analyze the binding properties of TaRXR, we expressed TaRXR as a GST-fusion protein (GST-TaRXR) in bacteria which was then purified as a GST-fusion protein and used directly for binding studies or cleaved by thrombin and eluted as TaRXR. The binding of $^3$H-labelled 9-*cis*-RA or $^3$H-labelled all-*trans*-RA was determined by measuring total bound radioactivity and the radioactivity displaceable by 200 fold excess of nonradioactive competitors. Consistent with the high conservation of the LBD, the experiments showed that TaRXR prepared as thrombin cleaved TaRXR or GST-TaRXR binds 9-*cis*-RA with high affinity and specificity (Figs. 2A and 2B). The 9-*cis*-RA binding assay showed high affinity binding to GST-TaRXR with a saturation plateau from 5 nM to 10 nM (Fig. 2C). In contrast, all-*trans*-retinoic acid did not show high affinity binding to TaRXR of GST-TaRXR.

## *9-cis*-retinoic acid induces malic enzyme gene expression at nanomolar concentrations

Next, we searched whether 9-*cis*-RA has observable biological effects on *T. adharens* at nanomolar concentrations. We hypothesized that TaRXR is likely to be involved in the regulation of metabolic events. In vertebrates, RXR is a dimerization partner of TR and together these two NRs are regulating a wide range of metabolic pathways. We, therefore, searched for an orthologue of vertebrate L-malate-NADP$^+$ oxidoreductase (EC 1.1.1.40) in *T. adhaerens* genome since this enzyme is an established reporter of the state of thyroid hormone dependent regulation (see 'Discussion').

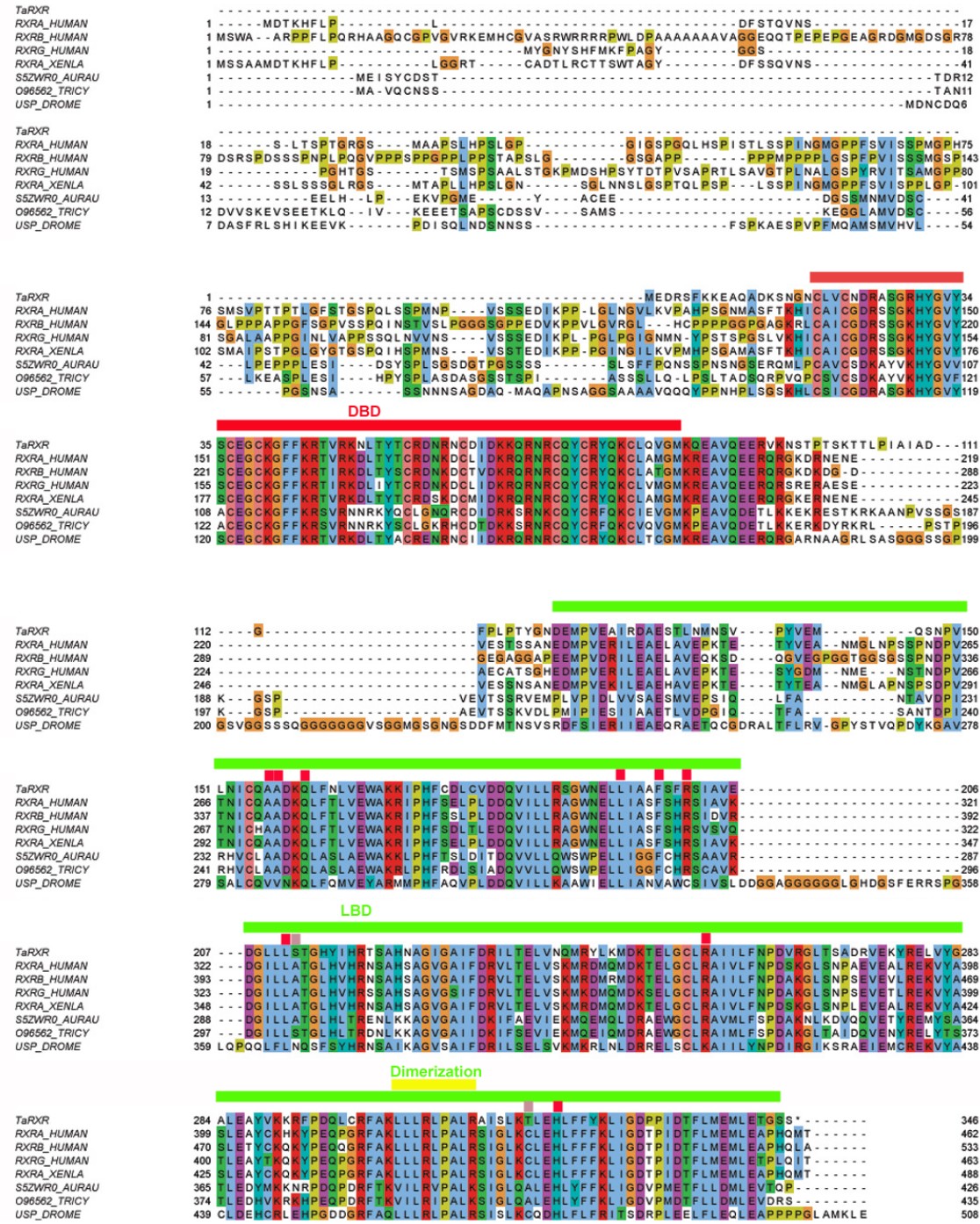

**Figure 1** **Multiple sequence alignment of selected metazoan homologues of RXR compared with TaRXR.** Aligned with ClustalO, amino acid residue types colored according to Clustal scheme in Jalview, * indicates DBD footprint residues, # LBD footprint residues. Black box shows the DBD, red box represents the LBD. Sequences from top to bottom (organism, identifier): *Trichoplax adhaerens*, TaRXR ID 53515; *Homo sapiens*, sp|P19793|RXRA_HUMAN; *Homo sapiens*, sp|P28702|RXRB_HUMAN; *Homo sapiens*, sp|P48443|RXRG_HUMAN; *Xenopus laevis*, RXR alpha, sp|P51128|RXRA_XENLA; *Aurelia aurita*, RXR, tr|S5ZWR0|S5ZWR0_AURAU Retinoid X receptor; *Tripedalia cystophora*, RXR, tr|O96562|O96562_TRICY Retinoic acid X receptor; *Drosophila melanogaster*, USP, sp|P20153|USP_DROME. DNA binding domain (DBD, red line), Ligand binding domain (LBD, green line), dimerization domain (yellow line) and amino acid residues critical for 9-*cis*-RA binding (conserved—red rectangles, not conserved—pink rectangles) are indicated. Readers with specific color preferences may download the compared sequences (File S1) and create the Clustal scheme with different color specifications using the Jalview program (http://www.jalview.org/).

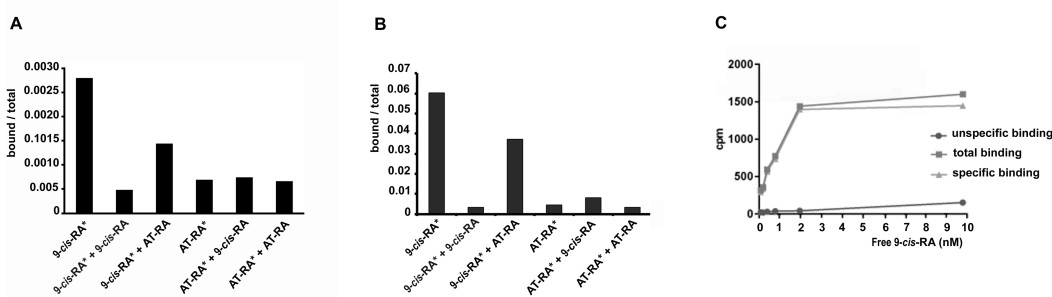

**Figure 2** **Binding of retinoic acids to TaRXR.** (A) Single point analysis of binding preference of
*T. adhaerens* RXR (thrombin cleaved) to $^3$H-labelled 9-*cis*-RA over all-*trans*-RA. Radioactive 9-*cis*-RA
(9-*cis*-RA*) binds at a concentration of 4 nM to 200 nanograms of *T. adhaerens* RXR. 200-fold excess
of unlabeled 9-*cis*-RA displaces more than 80% of labeled 9-*cis*-RA from binding to *T. adhaerens* RXR
(9-*cis*-RA* + 9-*cis*-RA) while the same molar excess of all-*trans*-RA (9-*cis*-RA*+ AT-RA) which is likely
to contain approximately 1% spontaneously isomerized 9-*cis*- RA, competes away less than 50 % of bound
$^3$H-labeled 9-*cis*-RA. Radioactive $^3$H-labeld all-*trans*-RA (AT-RA*) at identical conditions binds only
slightly more than the observed non-specific binding. This interaction is not displaced by the excess of
non-labeled 9-*cis*-RA (AT-RA* + 9-*cis*-RA) nor non-labeled all-*trans*-RA (AT-RA* + AT-RA). Results
are expressed as a ratio of the radioactivity bound to TaRXR/total radioactivity used for the binding at
the given condition. (B) Analysis of binding properties of *T. adhaerens* RXR (in the form of GST-TaRXR)
to $^3$H-labelled 9-*cis*-RA and $^3$H-labelled all-*trans*-RA. The experiment differs from the experiment
shown in A in 5-fold greater amount of radioactive all-*trans*-RA (and therefore only 40-fold excess of
non-radioactive competitors). The experiment shows identical binding properties of GST-TaRXR as
those observed with thrombin cleaved TaRXR. (C) Kinetic analysis of binding of $^3$H-labeled 9-*cis*-RA to
*T. adhaerens* RXR prepared as GST-fusion protein (GST-TaRXR). The plateau is reached at around 3 to
$5 \times 10^{-9}$ M.

The sequence of the *T. adhaerens* likely orthologue of vertebrate L-malate-NADP$^+$
oxidoreductase was retrieved from the *Trichoplax* genomic database together with its
presumed promoter based on the predicted sequence (File S3).

Droplet digital PCR showed an increased transcription of the predicted L-malate-
NADP$^+$ oxidoreductase gene after incubation of *T. adhaerens* with 9-*cis*-RA, but not with
all-*trans*-RA (Fig. 3). In repeated experiments, we observed that the level of induction
was higher at 9-*cis*-RA concentrations in the range of 1 to 10 nM, than above 10 nM. We
also noticed that the level of the induction slightly varied based on the actual *T. adhaerens*
cultures and the algal food composition of the *T. adhaerens* cultures.

## Changes in the culture environment alter the expression pattern of the nuclear receptor complement in *T. adhaerens*

From the experience we gained by culturing *T. adhaerens*, as well as from the previous
experiments we knew that the culture conditions could dramatically influence phenotype.
Having the possible developmental functions of the ancestral NRs in mind, we raised the
question whether the expression patterns of the NRs reflect changes in phenotype.

Firstly, we assayed the relative expression of RXR against all three other NRs in small
versus big animals (<0.5 mm or >1 mm). The relative proportion of & expression compared
to the remaining NRs was found to be higher in big animals (33%) than in small animals
(24%). The treatment by 3.3 nM 9-*cis*-RA led to a dramatic increase of the relative

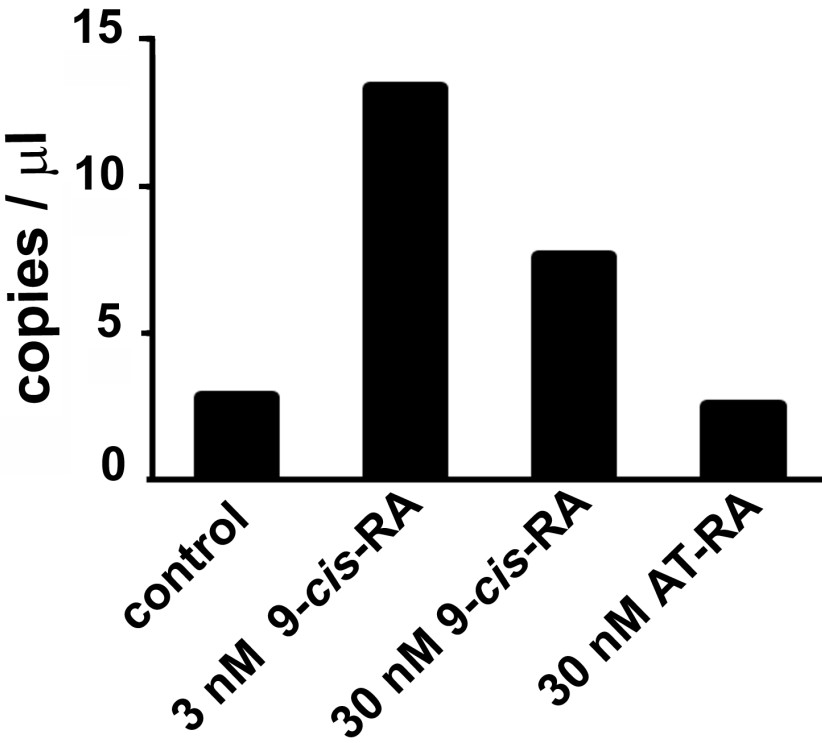

**Figure 3** **The effect of 9-*cis*-RA on the expression of the *T. adhaerens* closest putative homologue and likely orthologue of L-malate-NADP$^+$ oxidoreductase (EC1.1.1.40).** Ten to fifteen animals were cultured in the dark overnight with indicated ligands or in medium containing only the solvent used for ligand solutions. Total RNA and cDNA were prepared using identical conditions and diluted to the same working concentration suitable for ddPCR. In repeated experiments, incubation with 3 nM 9-*cis*-RA induced expression of the putative *T. adhaerens* L-malate-NADP$^+$ oxidoreductase more than four times. Incubation with 30 nM 9-*cis*-RA induced enzyme expression also, but to a lesser extent and 30 nM all-*trans*-RA (AT-RA) did not upregulate the expression of the predicted L-malate-NADP$^+$ oxidoreductase.

expression of RXR in comparison to the rest of the NR complement (51%), indicating that phenotypic changes are connected with differential expression of NRs and that 9-*cis*- RA affects the expression of RXR.

In order to see the effect of 9-*cis*-RA on all NRs, we sampled and extracted RNA from cultures containing the same number of big and small animals treated with different concentrations of 9-*cis*-RA. The experimental cultures were started from the same original cultures and during incubation were fed with *Chlorella sp.* only since this algal food showed to have the least effect on *T. adhaerens* cultures. All four *T. adhaerens* NRs were quantified by either qRT-PCR or ddPCR.

Analysis of NR expression pattern in animals incubated with different concentrations of 9-*cis*-RA, revealed a relative increase in RXR expression at low nanomolar concentrations (<10 nM) in repeated experiments. In contrast, further increase of 9-*cis*-RA resulted in smaller changes compared to the expression pattern of NRs in control animals or even reverted the values observed in low nanomolar conditions (Fig. 4).
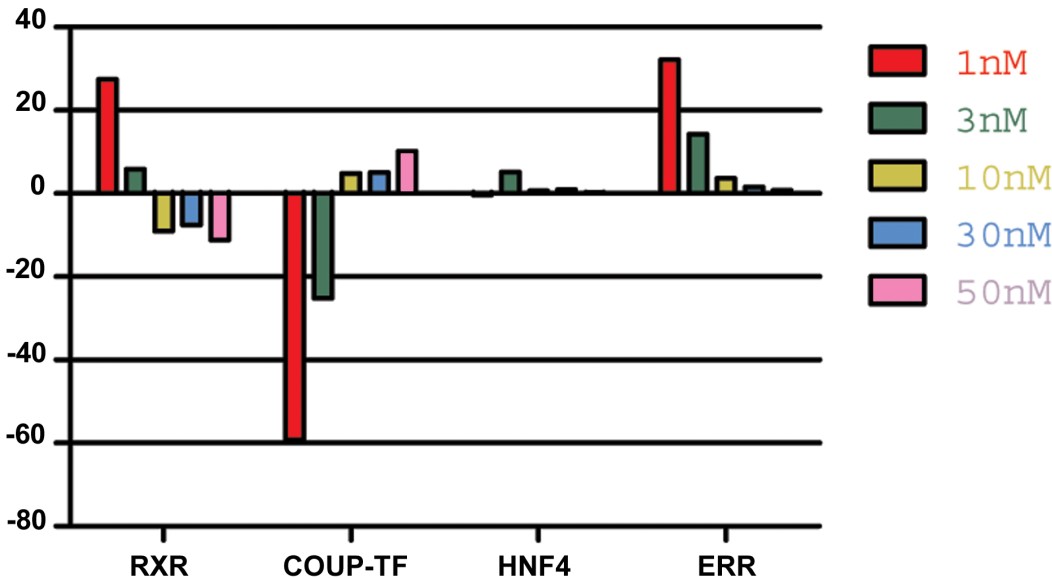

**Figure 4** **The effect of *9-cis*-RA on the expression of *T adhaerens* NRs.** A representative experiment of the expression of *T. adhaerens* NRs in animals exposed to various concentrations of 9-*cis*-RA expressed as a ratio of obtained values compared to the control using ddPCR. One and 3 nM 9-*cis*-RA upregulate RXR and ERR, but downregulate COUP-TF. The expression of *T. adhaerens* HNF4 is not affected by 9-*cis*-RA. The effect of the exposure to 9-*cis*-RA is stronger in the case of 1 nM 9-*cis*-RA compared to 3 nM 9-*cis*-RA. The exposure to 30 nM, as well as 50 nM concentrations of 9-*cis*-RA reverse the effect of 9-*cis*-RA on the expression of RXR and COUP-TF, but do not influence the expression of ERR. The level of expression of HNF4 is not changed by exposure of *T. adhaerens* to various concentrations of 9-*cis*-RA. The data suggest that a network sensitive to nanomolar concentrations of 9-*cis*-RA at the expressional level is formed by RXR, COUP-TF and ERR. All four NRs have conserved P-box (regions responsible for binding to response elements (RE) in gene promoters) and are likely to bind overlapping REs and to form a functional network.

## Food composition dramatically changes the phenotype and the reproduction rate of *T. adhaerens*

*T. adhaerens* retrieved from laboratory aquariums used for the stock cultures were relatively similar in appearance and included small round animals containing approximately 50 cells and grew to animals with an approximate diameter of 0.2 mm and rarely were bigger. Their rate of multiplication when transferred to Petri dishes was doubling in one month or even one week, depending on whether the glass was covered by microbial and algal films established during culturing in aquariums. We attempted to use several defined algae as artificial food. They included *Pyrrenomonas helgolandii*, *Picocystis salinarium*, *Tetraselmis subcoriformis*, *Rhodomonas salina*, *Phaeodactylum tricornutum*, *Porphyridium cruentum* and *Chlorella sp*. Individual subcultures of *T. adhaerens* differed in the rate of propagation and appearance as well as colors that were varying from greenish to brown and reddish taints depending on the food that was used as singular species food or mixtures (Fig. 5). Also, contaminants from the original algal food, which prevailed in some cultures, had an influence on *T. adhaerens* growth and behavior. In controlled experiments, it became clear that some food components or their metabolites are influencing growth and appearance of *T. adherens* more than food availability. When *T. adhaerens* were fed with

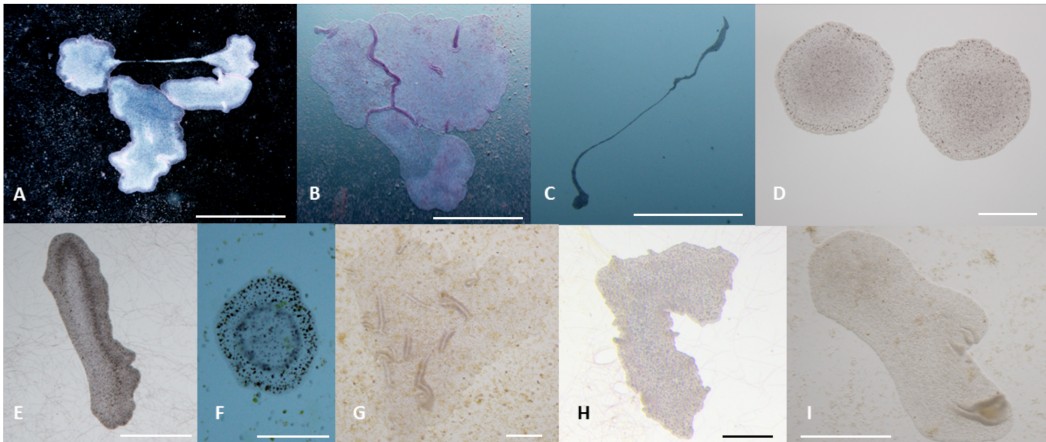

**Figure 5 Phenotypes of *T. adhaerens* change at various feeding conditions.** *T. adhaerens* acquires various body shapes in individual cultures dependent on food availability and composition. At conditions maintained in stable and biologically equilibrated stock aquariums, *T. adhaerens* is usually small and pale with diameter varying from 50 μm to 400 μm while cultures with added algae contain large flat animals with diameter reaching up to 1 mm (A and B). In some cultures, animals grow as long stretching structures, reaching a length exceeding one or even several centimeters (C). The algal food makes the animals greenish, reddish, rusty or brown with variable proportion of prominent dark cells. Animal shapes also vary from flat and round with smooth circumference, to curved or ruffled circumference or animals with long projections. Bars represent 1 mm in (A, B, I), 1 cm in C, 250 μm in (D), 500 μm in (E and H), 200 μm in (F) , and 100 μm in (G).

equal amounts of algal cells (although they differed in size and expected digestibility), the addition of algae containing red pigments—Cryptophytes (*Pyrrenomonas helgolandii* and *Rhodomonas salina*) or Rhodophyta (*Porphyridium cruentum*)—had a strong positive effect on *T. adhaerens* growth (Fig. 6), especially in combination with the green algae *Chlorella sp.* (Fig. 6).

Furthermore, the addition of *Porphyridium cruentum* to *Chlorella sp.* resulted in a significant change in circularity, while feeding *T. adhaerens* with 'triple food' containing *Chlorella*, *Rhodomonas* and *Porphyridium* showed the most pronounced effect. Culturing *T. adhaerens* on either of the single foods showed similar isoperimetric values (Fig. 7).

## 9-*cis*-RA interferes with *T. adhaerens* growth response to specific algal food

In order to see the effect of 9-*cis*-RA on *T. adhaerens* in cultures, we exposed cultures kept in a naturally established laboratory microenvironment or fed by specific algal foods to 3, 5 and 10 nM 9-*cis*-RA. The slowly growing cultures kept in naturally established laboratory microenvironment did not show any gross morphological changes even in 10 nM 9-*cis*-RA during the period of one week. Contrary to that, cultures fed with mixed algal food incubated in the presence of 3 and 5 nM 9-*cis*-RA ceased propagation and most animals developed a balloon-like phenotype, and later darkened and decomposed.

For controlled experiments, cultures fed by *P. cruentum* or *Chlorella sp.* were incubated in the presence of vehicle (DMSO/ethanol) or vehicle containing 9-*cis*-RA at 3.3 nM final concentration. After 24 h of incubation in the dark, control cultures fed by *P. cruentum*

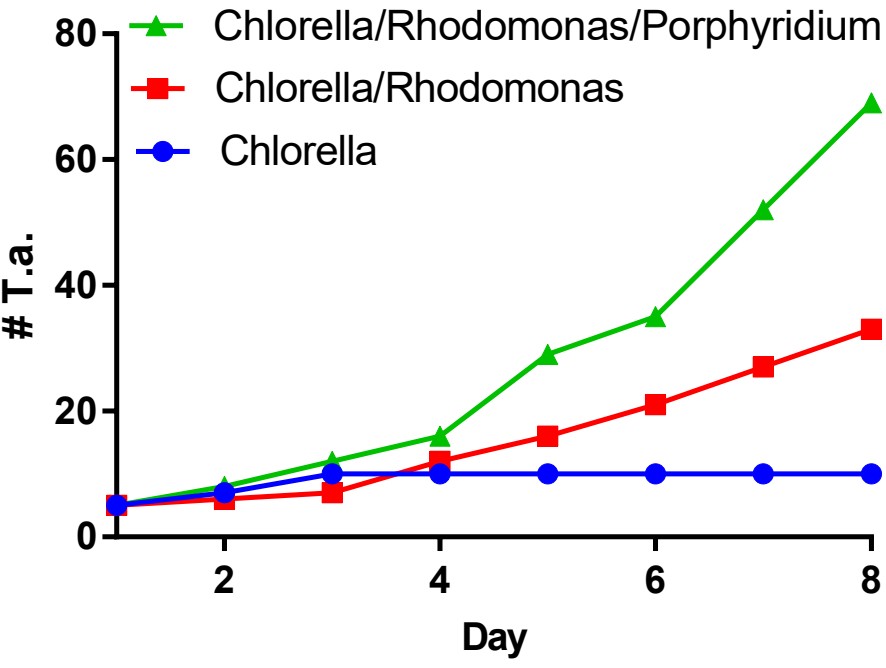

**Figure 6** **Propagation of *T. adhaerens* depends on algal food composition.** Three cultures of five large animals in each were established and fed with the same number of algal cells consisting of *Chlorella sp.*, *Chlorella sp.* and *Rhodomonas salina* and *Chlorella sp.*, *Rhodomonas salina* and *Porphyridium cruentum*. While the culture fed with *Chlorella sp.* only doubled in the number of animals within a period of one week, cultures with red pigment containing algae multiplied more than five times and 10 times within the same time period.

propagated normally while animals fed by *P. cruentum* and incubated with 9-*cis*-RA decreased their area and perimeter (Fig. 8A). At 72 h of incubation, all animals fed by *P. cruentum* and treated by 3.3 nM 9-*cis*-RA developed the balloon-like phenotype and none of them survived 90 h of exposure to 9-*cis*-RA (Fig. 8B and Fig. S2). Animals transferred from stationary cultures grown in a naturally established laboratory microenvironment and subsequently fed by *Chlorella sp.* suffered initial loses at 24 h of incubation despite that their appearance seemed to be normal and well adopted to the new culture condition at time 0 (regarding feeding with algal food and immediately prior to addition of vehicle or 9-*cis*-RA to the culture and 6 h after the transfer from the parent cultures). Animals that survived the transfer and adopted to feeding by *Chlorella sp.*, were not inhibited by exposure to 3.3 nM 9-*cis*-RA for 24 h (Fig. 8C) and even showed a slight statistically not significant increase in their area and perimeter. Nevertheless, the isoperimetric values of animals incubated for 24 h with 9-cis-RA showed a significant increase indicating a decrease of growth or exhaustion of peripheral area, that is likely to contain stem cells that further differentiate into the specialized cell types (*Jakob et al., 2004*). In contrary to animals fed by *P. cruentum*, exposure to 9-*cis*-RA was not associated with the development of the balloon-like phenotype and animals survived more than 250 h (Fig. S3). In contrast to control animals which started to proliferate after 100 h, animals exposed to 9-*cis*-RA did not proliferate between 90 and 280 h of subsequent culture (Fig. 8D) suggesting that

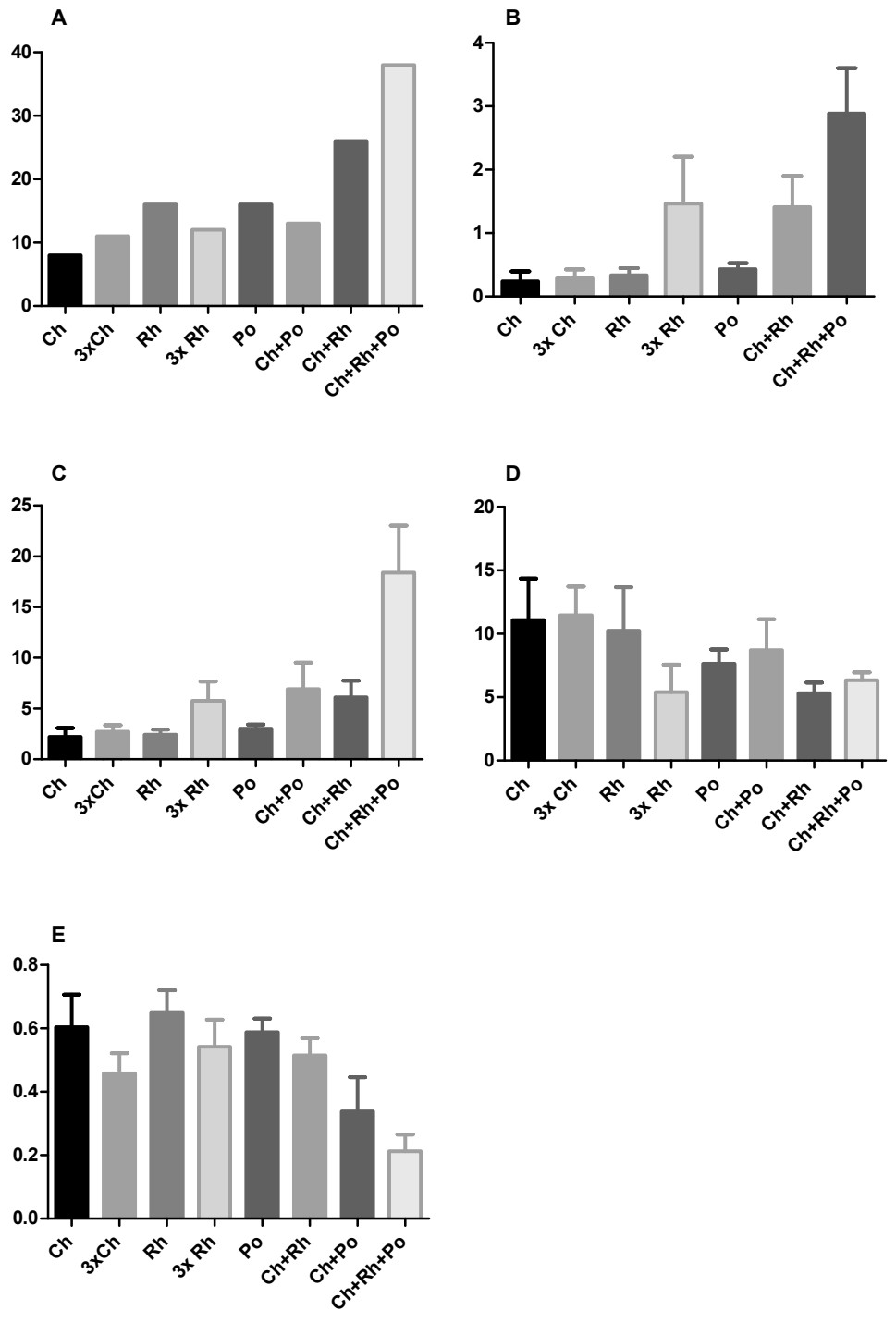

**Figure 7 The effect of algal food composition on *T. adhaerens* growth and appearance.** *T. adhaerens* was cultured similarly as shown in Fig. 6 and all animals were photographed and analyzed using ImageJ program for their number (A), mean area (B), mean perimeter (C), mean perimeter/area ratio (D) and mean isometric quotient (E) (continued on next page...)

**Figure 7 (...continued)**
after one week. Ch—stands for feeding with *Chlorella sp.*, Rh—*Rhodomonas salina*, Po—*Porphyridium cruentum*, and their combinations. 3Rh stands for a culture with three times higher concentration of *Rhodomonas salina* and 3Ch for three times higher concentration of *Chlorella sp.* (A) shows that addition of *Rhodomonas salina* (Ch + Rh) greatly increases the number of animals observed after one week of culture. This effect is even more pronounced in cultures containing all three algae, while three times bigger concentration of only one type of algae (Ch and Rh) has little or no effect. This is even more pronounced when the area and perimeter are determined (B and C). Determination of the isoperimetric quotient in individual cultures indicates that cultures with *Rhodomonas salina* have a significantly smaller ratio, suggesting higher proliferative rate of structures at the animal circumference (E). Bars represent 95% confidence interval. Raw data are provided as Files S5 and S6.

9-*cis*-RA interferes with animal response to specific food and processes necessary for animal growth and propagation. The growth arrest of *T. adhaerens* caused by 9-*cis*-RA was reverted by addition of *Porphyridium cruentum* indicating that a specific food constituent rather than food availability interferes with 9-*cis*-RA regulatory potential (Fig. S4).

## DISCUSSION

### *T. adhaerens* is probably the closest living species to basal metazoans with only four NRs

*Trichoplax adhaerens* is an especially interesting species from an evolutionary perspective. It shows the most primitive metazoan planar body arrangement known with a simple dorsal-ventral polarity, the establishment of which is one of the most ancient events in evolution of animal symmetries (*Smith et al., 1995*; *Stein & Stevens, 2014*). The Placozoa are disposed with only a few (probably six) morphologically recognizable cell types (*Jakob et al., 2004*; *Smith et al., 2014*).

In strong contrast to this, the *T. adhaerens* genome shows larger blocks of conserved synteny relative to the human genome than flies or nematodes (*Srivastava et al., 2008*). Genome analyses indicate that Placozoa are basal relative to Bilateria as well as all other diploblastic phyla (*Schierwater et al., 2009*), but all kinds of different views are also discussed (reviewed in *Schierwater et al., 2016*).

In concordance with this, its genome contains four (*Srivastava et al., 2008*) rather than 17 NRs, which can be found in the cnidarian *Nematostella vectensis* (*Reitzel & Tarrant, 2009*). Even though it has been proposed that Placozoa lost representatives of NR6 (SF1/GCNF), TR2/TR4 of the NR2 subfamily and invertebrate specific nuclear receptors (INR, clade of invertebrate-only nuclear receptors with no standard nomenclature) NR1/NR4 (*Bridgham et al., 2010*). The reasoning in this direction depends on the assumed phylogenic position of the phylum Placozoa.

The four NRs found in the genome of *T. adhaerens* are relatively highly related to their vertebrate counterparts, RXR (NR2B), HNF4 (NR2A), COUP-TF (NR2F) and ERR (NR3B) (*Srivastava et al., 2008*). Among them, *T. adhaerens* RXR and HNF4 show the highest degree of identity in protein sequence and the relatedness of *T. adhaerens* RXR (TaRXR) to human RXR is similar to that of *Tripedalia cystophora* RXR (jRXR) (*Kostrouch et al., 1998*), which has also been shown to bind 9-*cis*-RA at nanomolar concentrations. These results suggest that TaRXR is structurally and also functionally very closely related

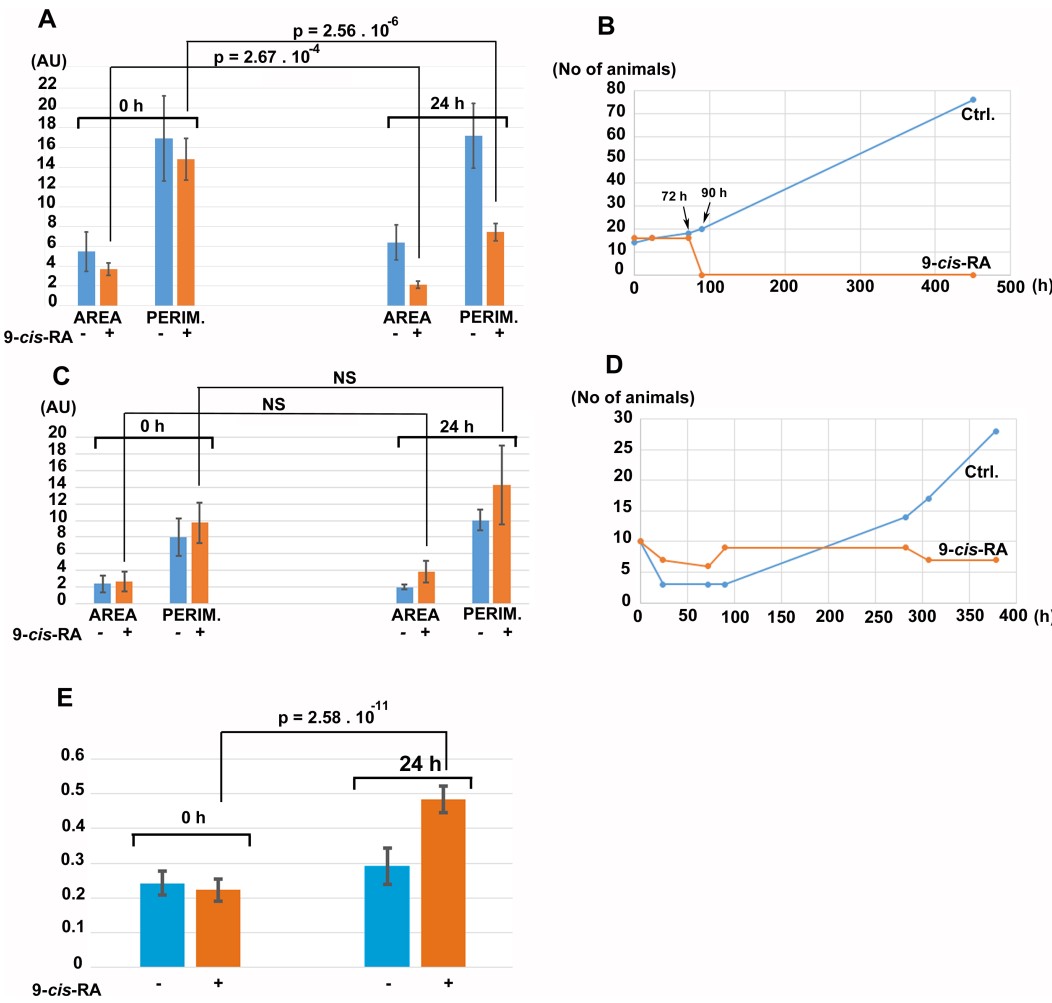

**Figure 8** The effect of 9-*cis*-RA on growth of *T. adhaerens* fed by *Porphyridium cruentum* or *Chlorella sp.* (A) shows the comparison of the total area and total perimeter of control *T. adhaerens* and *T. adhaerens* treated by 9-*cis*-RA for 24 h expressed as arbitrary units derived from pixel measurements obtained at 24 h and compared to values obtained immediately prior to incubation. The data indicates that animals incubated in 3.3 nM 9-*cis*-RA decreased their area and perimeter to approximately 50% in comparison to control animals. (B) shows the development of cultures over a three-week period. The animals treated by 9-*cis*-RA developed a balloon-like phenotype at 72 h of incubation and died at 90 h of incubation. (C and D) show the data obtained with *T. adhaerens* originating from cultures fed by naturally established biofilms in laboraotory aquariums, fed by *Chlorella sp*. and treated similarly as shown in (A and B). (E) shows the analysis of circularity of animals presented in (A) and documents that animals treated by 9-*cis*-RA increased their circularity already at 24 h of exposure suggesting arrest of growth of animal peripheral tissues.

to its vertebrate counterparts, most probably representing the most basal liganded NR of all Metazoa known today.

## *T. adhaerens* RXR binds preferentially 9-*cis*-RA

By using a radioactively labelled ligand, we could demonstrate that the RXR orthologue in *T. adhaerens* binds 9-*cis*-RA with very high affinity and shows a strong binding preference

to 9-*cis*-RA over all-*trans*-RA similarly, as is the case in vertebrate RXRs (*Allenby et al., 1993*) and the cnidarian TcRXR (*Kostrouch et al., 1998*).

### 9-*cis*-RA affects the expression of a *T. adhaerens* orthologue of a conserved metabolically active enzyme, L-malate-NADP$^+$ oxidoreductase

A biological role of 9-*cis*-RA binding with high affinity to the placozoan RXR receptor is supported by our *in vivo* experiments. In the search for genes that may be under the regulation of TaRXR, we identified a putative orthologue of vertebrate L-malate-NADP$^+$ oxidoreductase (EC 1.1.1.40) and analyzed the effect of 9-*cis*-RA or all-*trans*-RA on its expression. In agreement with our binding experiments, we observed induction of this enzyme's expression at low nanomolar concentrations of 9-cis-RA (1 to 10 nM). Interestingly, higher concentrations of 9-*cis*-RA (30 nM) had a smaller effect on expression and all-*trans*-RA had no effect up to 30 nM concentrations. A plausible explanation for this could be that 9-*cis*-RA also acts as a ligand for other *T. adhaerens* NRs which may have an opposite effect on the expression of L-malate-NADP$^+$ oxidoreductase. Furthermore, 9-*cis*-RA could act as a supranatural ligand and the continuous occupation of TaRXR by this high affinity ligand may interfere with the normal function of the receptor within the transcription initiation machinery.

In mammals, regulation of malic enzyme expression is mediated by a thyroid hormone receptor (TR)—RXR heterodimer (*Dozin, Cahnmann & Nikodem, 1985*; *Dozin, Magnuson & Nikodem, 1985*; *Petty et al., 1989*; *Petty et al., 1990*). By showing a 9-*cis*-RA dependent change in the expression of the likely placozoan malic enzyme orthologue *in vivo*, we provide indirect evidence of a conserved RXR mediated regulation of gene expression. Although the expression of L-malate-NADP$^+$ oxidoreductase in mammals is usually used as a factor reflecting regulation by thyroid hormone (*Dozin, Magnuson & Nikodem, 1986*), it has also been shown that its cell-type associated differences depend on the expression level of RXR alpha (*Hillgartner, Chen & Goodridge, 1992*; *Fang & Hillgartner, 2000*) suggesting that regulation by RXR has been conserved throughout metazoan evolution while additional regulation via thyroid hormone represents an innovation of Bilateria (*Wu, Niles & LoVerde, 2007*).

### NRs form a network responding to 9-*cis*-RA

Since autoregulation and cross-regulation of NRs by their specific ligands is well documented for a large number of nuclear receptors (*Tata, 1994*), we searched if 9-*cis*-RA affects the expression of TaRXR mRNA relative to the other *T. adhaerens* NRs. Our *in vivo* experiments showed not only effects on specific gene expression in response to very low concentrations of 9-*cis*-RA (at 1 or 3 nM), but also an additional dose-dependent reverse effect of higher concentrations. This is likely to be in line with our binding experiments that suggested the possibility of an additional binding site or sites with higher capacity and lower affinity. We also cannot rule out that higher concentrations of 9-*cis*-RA affect some of the three remaining *T. adhaerens* NRs. Nevertheless, an inhibitory effect of 9-*cis*-RA on the expression of its cognate receptor at the protein level (through protein degradation) was reported (*Nomura et al., 1999*).
Although it is not clear if 9-*cis*-RA is the natural ligand for RXRs (*Wolf, 2006*; *Ruhl et al., 2015*) conserved in all metazoan phyla studied to date, we show not only that 9-*cis*-RA binds TaRXR with nanomolar affinity but also positively regulates its expression, which resembles auto-activation of several NRs in vertebrates (e.g., ER and TR (*Tata, 1994*; *Bagamasbad & Denver, 2011*)). Furthermore, three out of four NRs constituting the NR complement in *T. adhaerens* respond to treatment by 9-*cis*- RA at transcriptional level. Two NRs, RXR itself and ERR, respond positively to nanomolar concentrations of 9-*cis*-RA, while COUP-TF, which often acts as an inhibitor of specific gene expression (*Tran et al., 1992*), is regulated negatively by 9-*cis*-RA. COUP-TF was recently shown to be inactivated by small hydrophobic molecules (*Le Guevel et al., 2017*). The regulatory connections of *T. adhaerens* NRs places the autoregulation and cross-regulation of NRs to the base of metazoan evolution. The proposed regulatory network of *T. adhaerens* NRs is schematically represented in Fig. S5.

## Food composition rather than quantity affects phenotype of *T. adhaerens*

At first glance, *T. adhaerens* seems to benefit from any source of biological material on surfaces that can be digested and absorbed by its digestive system (e.g., aquarium microorganisms and detritus). Feeding with certain live microorganisms in laboratory cultures, however, dramatically changes the dynamics of *T. adhaerens* cultures, such as shape, size, color, body transparency, growth and divisions of the animal. For example, we observed poor growth and reproduction rates of *T. adhaerens* fed solely on *Chlorella sp.* even at a relatively high density. In contrast, cultures fed with red pigment containing *Rhodomonas salina* showed much faster proliferation and led, in part, to the formation of giant animals, seeming to halt their division. Despite *Porphyridium cruentum* containing similar pigments as *Rhodomonas*, such as phycoerythrin, cultures grown with *Porphyridium* as the main nutrient source did not show phenotypical abnormalities but the addition of it to a culture with *Chlorella* and *Rhodomonas* resulted in an additive effect on reproduction rate.

Even though the growth of *T. adhaerens* seems to follow a simple program, it is likely to require strict regulatory mechanisms. Formation of specific cellular types is connected with phenotypic appearance of animals possessing larger proportions of certain cells, e.g., upper epithelium in balloon-like animals or larger proportion of peripheral regions containing stem cell-like cells in narrow or prolonged animals. Analysis of circularity as a measure of location specific cellular proliferation is in concordance with the observed culture characteristics and shows that lower isoperimetric values (less 'roundness') indicate higher reproduction rates.

Our experiments provide evidence that food composition is more important for *T. adhaerens* growth and propagation than its quantity, which is in line with the recent finding of phosphate and nitrate playing important roles determining the distribution of placozoans around the globe (*Paknia & Schierwater, 2015*). It indicates that food constituents, especially those present in the algae containing phycobilin based red pigments like *Rhodomonas salina* and *Porphyridium cruentum* might possess hormone-like molecules or molecules resulting in hormone-like metabolites in *T. adhaerens* that act through the

NR complement and, indeed, analysis of NRs in differently sized animals indicates impact of food composition on NR expression.

The high sensitivity of *T. adhaerens* to 9-*cis*-RA reflected by the transcriptional response to low nanomolar concentrations of 9-*cis*-RA but not all-*trans*-RA and the interference of 3.3 nM 9-*cis*-RA with the animal response to feeding together with the high affinity binding of 9-*cis*-RA by TaRXR suggests that the response of *T. adhaerens* to 9-*cis*-RA is mediated by TaRXR. It cannot be excluded that other *T. adhaerens* NRs, especially TaCOUP-TF and possibly also TaHNF4 may also be affected by 9-*cis*-RA. It seems possible that 9-*cis*-RA or similarly shaped molecules may be present in *T. adhaerens* food or can be formed from retinoids and other molecular components of food. Our data indicate that the sensitivity of *T. adhaerens* to 9-*cis*-RA depends on the actual feeding conditions and animal growth. There are several possible scenarios that may explain the high sensitivity of *T. adhaerens* to 9-*cis*-RA. The activation of RXR by 9-*cis*-RA or similar compounds has been documented in vertebrates (*Allenby et al., 1993*; *Ruhl et al., 2015*; *De Lera, Krezel & Ruhl, 2016*). The observation of 9-*cis*-RA induced growth arrest is similar as data reported on mammalian cells (e.g., *Wente et al., 2007*) however the concentration of 9-*cis*-RA used in our experiments is approximately 30 to 3,000 times lower than the levels reported in most mammalian systems. It seems likely that very low concentrations of natural ligands including 9-*cis*-RA or similarly shaped molecules or other molecules present in the algal food or produced from algal food as metabolites in *T. adhaerens* regulate the gene expression via RXR in *T. adhaerens*. This may be connected with *T. adhaerens* strong response to light exposure visible as coordinated relocations of animals inside laboratory culture containers and a strong influence of annual seasons on *T. adhaerens* propagation rates observed in laboratories localized in temperate geographical zones. The 9-*cis* conformation of RA is not only sensitive to light exposure with its reversal to all-trans conformation but it can also be formed by specific UV irradiation from all-trans conformation up to 10% as shown by Dr. Hans Cahnmann (*Cahnmann, 1995*).

Chlorophyll hydrophobic side chain which anchors the molecule to the chloroplast thylakoid membrane is metabolized to phytol that was shown to act as an RXR agonist (*Kitareewan et al., 1996*). Other molecules called rexinoids, which often contain aromatic rings in their structure act as RXR agonists or antagonists (*Dawson & Xia, 2012*). Lately, another group of ligands called organotins was shown to affect regulation by RXR (*Le Maire et al., 2009*). It has been proposed that RXRs can bind a larger group of polyunsaturated fatty acids (docosahexaenoic acid and arachidonic acid) and act as their sensors (*De Urquiza et al., 2000*; *Lengqvist et al., 2004*).

When viewed together, our work shows the presence of 9-*cis*- RA binding RXR in Placozoa and argues for the existence of ligand regulated NRs at the base of metazoan evolution. Our observations suggest the existence an endocrine-like regulatory network of NRs in *T. adhaerens* (schematically represented in Fig. S5). Endocrine, hormone-receptor regulations involving NRs may be viewed as specialized, very powerful yet not prevailing regulations transmitted by NRs. Increasingly larger numbers of non-hormonal ligands originating in environment, food or metabolism are emerging as regulatory molecules of NRs (*Holzer, Markov & Laudet, 2017*). Our data suggest that non-hormonal, environment

and food derived ligands are likely to be the first or very early ligands regulating the metazoan response to food availability and orchestrating growth of basal metazoans and necessary differentiation to specialized cell types. In this sense, NRs in *T. adhaerens* represent an endocrine-like system of ancestor NRs.

This work suggests that ligand regulated RXR is involved in the coordination of animal growth and development throughout the metazoan evolution. This also suggests that the regulation by liganded NRs evolved as an evolutionary need connected with heterotrophy and multicellularity.

In fact, despite fragments of NR domains being found in prokaryotes, no single full sized NR has been discovered in bacteria or archaea and the closest known relatives to metazoans, unicellular and colonial Choanoflagellates, lack nuclear receptors, as well as genes of several other regulatory pathways (*King et al., 2008*). On the other hand in fungi, the sister group of Holozoa, *Shalchian-Tabrizi et al., (2008)* transcription factors surprisingly similar to metazoan NRs evolved independently possibly for the regulation of metabolism and response to xenobiotics (*Thakur et al., 2008*; *Naar & Thakur, 2009*). Thus, the evolution of NRs seems to be associated with two key evolutionary features of metazoans: multicellularity and heterotrophy.

Ctenophores, a possible sister phylum to *Cnidaria*, do not contain classical NRs featuring both mechanistically critical domains of NRs, the DNA binding and ligand binding domains. Nevertheless, the ctenophore *Mnemiopsis* contains two orthologues of NR2A (HNF4) that lack the DNA binding domain (*Reitzel et al., 2011*).

Our observations of the exceptionally high sensitivity of *T. adhaerens* to 9-*cis*-RA imply the possibility that the originally very strong regulations mediated by NRs might have been softened or inhibited by additionally evolved mechanisms. To our knowledge, there are no reports of 100% lethal effects of exposure to low nanomolar levels of 9-*cis*-RA in any metazoan organism. It can be speculated that these mechanisms were likely to evolve to modulate 9-*cis*-RA's or similar ligand's regulatory potential further and might involve stronger regulations by heterodimerization partners of RXR and enzymatic or transport mechanisms regulating the availability of ligands in cells and tissues of more recent Metazoa.

In conclusion, the presence of functional nuclear receptors in *T. adhaerens* and their proposed regulatory network support the hypothesis of a basic regulatory mechanism by NRs, which may have been subspecialized with the appearance of new NRs in order to cope with new environmental and behavioral challenges during the course of early metazoan evolution and developmental regulatory needs of increasingly more complex metazoan species.

## ACKNOWLEDGEMENTS

The databases of NCBI (*NCBI Resource Coordinators, 2017*) and Joint Genome Institute of United States Department of Energy (http://jgi.doe.gov/) (*Nordberg et al., 2014*) provided bioinformatics support for this study.

### Funding

The main funding sources were: 1/the European Regional Development Fund "BIOCEV—Biotechnology and Biomedicine Centre of the Academy of Sciences and Charles University in Vestec" (CZ.1.05/1.1.00/02.0109) (The Start-Up Grant to the group Structure and Function of Cells in Their Normal State and in Pathology—Integrative Biology and Pathology (5.1.10)); 2/ The grant PRVOUK-P27/LF1/1 from the Charles University; 3/The grants SVV 260377/2017, SVV260257/2016, SVV260149/2015 and SVV 260023/2014 from the Charles University. This work was supported by the research project PRVOUK—Oncology P27, awarded by Charles University in Prague and by the project OPPK No. CZ.2.16/3.1.00/24024, awarded by European Fund for Regional Development (Prague & EU—We invest for your future). PROGRES Q26/LF1. This work was also supported by the Ministry of Education, Youth and Sports of CR within the LQ1604 National Sustainability Program II (Project BIOCEV-FAR) and by the project "BIOCEV" (CZ.1.05/1.1.00/02.0109). The work of Radek Kaňa was further supported by GACR 16-10088S and by the institutional projects Algatech Plus (MSMT LO1416) and Algamic (CZ 1.05/2.1.00/19.0392) provided by the Ministry of Education, Youth and Sports of the Czech Republic. For getting the project started Bernd Schierwater received support from the German Science Foundation (DEG Schi 277/27-1 and Schi 277/29-1). Authors received monetary support of the work reported in this publication from MediCentrum Praha. Zdenek Kostrouch and Marta Kostrouchová contributed with personal funds to this work. The funders (except authors) had no role in study design, data collection and analysis, decision to publish, or preparation of the manuscript.

### Grant Disclosures

The following grant information was disclosed by the authors:
European Regional Development Fund "BIOCEV—Biotechnology and Biomedicine Centre of the Academy of Sciences and Charles University in Vestec: CZ.1.05/1.1.00/02.0109.
Charles University: PRVOUK-P27/LF1/1, SVV 260377/2017, SVV260257/2016, SVV260149/2015, SVV 260023/2014.
PRVOUK—Oncology P27.
European Fund for Regional Development: OPPK No. CZ.2.16/3.1.00/24024.
Grant Agency of the Czech Republic: 16-10088S.
The Ministry of Education, Youth and Sports of the Czech Republic: CZ 1.05/2.1.00/19.0392.
German Science Foundation: DEG Schi 277/27-1 and Schi 277/29-1.

### Competing Interests

Marta Kostrouchová is an Academic Editor for PeerJ. Authors declare there are no competing interests.

### Author Contributions

- Jan Philipp Novotný, Ahmed Ali Chughtai, Markéta Kostrouchová, Veronika Kostrouchová and David Kostrouch conceived and designed the experiments, performed

the experiments, analyzed the data, wrote the paper, prepared figures and/or tables, reviewed drafts of the paper.

- Filip Kaššák conceived and designed the experiments, performed the experiments, analyzed the data, wrote the paper, reviewed drafts of the paper.
- Radek Kaňa conceived and designed the experiments, contributed reagents/materials/-analysis tools, wrote the paper, reviewed drafts of the paper.
- Bernd Schierwater analyzed the data, contributed reagents/materials/analysis tools, wrote the paper, reviewed drafts of the paper.
- Marta Kostrouchová and Zdenek Kostrouch conceived and designed the experiments, performed the experiments, analyzed the data, contributed reagents/materials/analysis tools, wrote the paper, prepared figures and/or tables, reviewed drafts of the paper, contributed with perosnal funds.

## DNA Deposition

The following information was supplied regarding the deposition of DNA sequences:

The sequence has been deposited in GenBank (accession number: MF805762). The raw data has been submitted as Supplementary Files.

## Data Availability

The raw data has been submitted as Supplementary Files.

## Supplemental Information

Supplemental information for this article can be found online at http://dx.doi.org/10.7717/peerj.3789#supplemental-information.

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
