# Peer review of "Trichoplax adhaerens reveals a network of nuclear receptors sensitive to 9-cis-retinoic acid at the base of metazoan evolution"

_PeerJ, doi:10.7717/peerj.3789_

## Round 0.1 · original submission · Major Revisions

Many thanks for your submission. I hope that you will find the reviewers' comments useful, and would greatly welcome your resubmission of this article. Kind regards, Chris

Reviewer 1 ·

Basic reporting

Review on the manuscript by Novotný et al. “Trichoplax adhaerens reveals an endocrine-like network sensitive to 9-cis-retinoic acid at the base of metazoan evolution”

In Srivastava et al. 2008, four different nuclear receptor (NR) family transcription factors were reported to be encoded in the genome of Trichoplax adhaerens, a basal animal that is morphologically very simple. The four Trichoplax family members, RXR (NR2B), HNF4 (NR2A), COUP (NR2F) and ERR (NR3B), show a high degree of similarity to their human homologs. NRs are normally defined as ligand-activated transcription factors that allow the regulation of target genes by small lipophilic molecules such as hormones (thyroid hormones, steroids), morphogens (retinoic acid) or dietary components (fatty acids).
In the present article, Novotný et al. explore the role of NRs in Trichoplax. First, they show that one of the four NRs of Trichoplax, the retinoid X receptor homolog (TaRXR), is highly conserved (about 66% identity to vertebrate homologs) and is able, like the human homolog, to bind to 9-cis-retinoic acid (9-cis-RA) in the low nanomolar range. To show this, they have used an assay with radioactive 9-cis-RA established earlier (e.g. used to describe the affinity of the RXR of the cnidarian Tripedalia cystophora by Kostrouch et al. 1998). Novotný et al. also show that 9-cis-RA increases the expression of a malic enzyme they had identified in Trichoplax. They suggest that this could reflect a natural regulation of this metabolic enzyme by activated TaRXR. Moreover, the authors observed a change of the expression pattern of three of the four NRs induced by 9-cis-RA. Apparently, the authors also observed a difference in the expression patterns of NRs when they compared animals of different sizes. In the last part of the manuscript, the authors report that the food source has an impact on the size and growth rate of Trichoplax. I have to admit that I did not quite understand why these observations have been added to the manuscript about NRs and the effect of 9-cis-RA on TaRXR. The observations on the effect of food on Trichoplax are not really connected with the rest of the report. Somehow, when I read the abstract, I thought that addition of 9-cis-RA to the culture would have changed the shape, size or behaviour of the animals – it almost goes without saying that this would be very interesting. Have such experiments been performed and no changes were observed? Do the authors think that components of the algae might be ligands for the NRs? Another possibility to link the two sections would have been to test for changes of the expression level of different NRs upon feeding the animal with different algae.

Overall, I find the observations presented in the manuscript interesting, but several aspects were not entirely clear to me. I therefore suggest that the authors re-analyze their data and carry out additional experiments before considering publication of their findings. Moreover, I do not agree with the authors that the results indicate the presence of an “endocrine-like network” in Trichoplax. Please note that the current title of the manuscript does not really make sense. The term “endocrine-like network probably refers to the possible network of NRs found in the study, but the term endocrine appears a bit far-fetched as it implies the presence of an endocrine system in the animal, for which the present study does not provide clear evidence.

Major points that need to be revised:

1) In the Introduction, the authors do not make sufficiently clear why they have studied the NRs of Trichoplax adhaerens in particular. For example, they write “that Life on Earth began 4.1 to 3.5 billion years ago (Bell et al. 2015) with the appearance of the first unicellular prokaryotic organisms that subsequently evolved, in part, to multicellular lifeforms forming the kingdom Metazoa that have specialized tissues for digestion, regulation of homeostasis, locomotion, perception, analysis of the environment and reproduction.” This is obviously not correct: animals are eukaryotic organisms. Several eukaryotic lineages are multicellular: animals, fungi, plants and others. Here it would be advisable to shortly explain what we know about the evolution of NRs and their ligands in animals (and other eukaryotes), for example as has been done in Kostrouchova & Kostrouch 2015, for example. Trichoplax with its relatively simple set of NRs represents an animal at the base of the animal kingdom and could therefore shed light on an early set of NRs in animals, if that was the aim of the study. I think that this point is currently more clearly outlined in the Discussion, a section of the manuscript written in a clearer fashion than the rest of the manuscript in my opinion.

2) Remarkably, Trichoplax has a set of NRs that are highly similar to human NRs, in particular TaRXR is highly conserved. The conservation is shown by a sequence alignment, but I did not understand why only few organisms are represented in the alignment (and in the tree). Why, for example, are model cnidarians like Nematostella or Hydra not included? Don’t they have the particular receptor? What about sponges and ctenophores? What about bilaterians? This should be explained and the data set, if necessary, expanded. I suggest to improve the representation of the sequence alignment by clearly indicating the domains (DBD & LBD) and residues involved in 9-cis-RA. If these residues were indicated, the few non-conserved residues do not need to be outlined in the text (line 294). Check for example Gutierrez-Mazariegos et al.. 2016. Evolutionary diversification of retinoic acid receptor ligand-binding pocket structure by molecular tinkering. R. Soc. open sci. 3:150484 (PMID: 27069642). It also seems that the authors are unaware of a study, in which the set of Trichoplax NRs was investigated already (Baker ME. 2008. Trichoplax, the simplest known animal, contains an estrogen-related receptor but no estrogen receptor: Implications for estrogen receptor evolution. Biochem Biophys Res Commun 375:623–627. PMID: 18722350). By the way, I did not understand in the manuscript (line 280) whether the known TaRXR sequence (Srivastava et al. 2008) was used or whether the authors generated a revised/improved sequence of the TaRXR. If so, what exactly was improved?
Furthermore, I suggest to omit the phylogenetic analysis. As far as I understood, the important point was to show that the 9-cis-RA binding site is conserved between man and Trichoplax. Probably there is a protein structure that can be used to discuss the conservation and binding site. A tree that is based only on a few sequences does not provide much information and might be biased, because only selected sequences have been used. I also suggest to omit Table 1, because a sequence comparison based on a few Blast search results does not provide much information. For both data sets, a more rigorous analysis would be necessary. Again, the authors should compare their results to the above mentioned study by Baker ME. 2008.

3) In the section about 9-cis-RA binding, the authors only state that 9-cis-RA binds with high affinity and specificity, but do not explain in the text nor in the figure legend how the experiments have been performed. Currently, the reader has to go to the Method section to find out that this was done using radioactive 9-cis-RA (how was radioactivity measured?) which was added to recombinant proteins expressed in E. coli. Why isn’t the purity of the proteins shown by SDS-PAGE, for example? Was TaRXR used as a GST fusion protein or was the cleaved protein used in the experiments shown? How does the affinity measured compare to the affinity of 9-cis-RA to the human (or others) protein reported in earlier studies? That “this was clearly observed in repeated experiments” (line 319) should not be stated in the Results, rather the figure legend should make clear that the experiments were carried out several (how often?) times, also by adding error bars.

4) The upregulation of the malic enzyme in Trichoplax has been demonstrated by quantitative PCR (Fig. 4). However, no control of a housekeeping gene like actin or another enzyme has been performed to make sure that really similar amounts have been used. This also concerns the experiments shown in Fig. 5 and experiments mentioned only in the text, where the authors report about different sizes of animals. Moreover, as no error bars are shown, it seems that the experiments have not been repeated sufficiently. What I did not understand was why the data in Fig. 5 are not given, as in Fig. 4, as copies/µl? What are the values on the y-axis in Fig. 5? Furthermore, the data shown in Fig. 5 suggest that the effect of 9-cis-RA on Trichoplax was strongest at 1 nM concentration. But why then were the experiments shown in Fig. 4 only carried out at 3 nM and at higher concentrations, where, according to the experiments shown in Fig. 5, rather opposing effects of 9-cis-RA have been observed?

5) The date shown in Fig. 5 are interpreted by the authors as an expression network induced by 9-cis-RA. It remained unclear to me, however, whether the authors imply that 9-cis-RA elicits its effect solely via binding to TaRXR, which in its activated form then activates the expression of its own gene and the genes of the other NRs, or whether they think that 9-cis-RA can also interact with the other NRs at nanomolar concentration. Binding experiments with 9-cis-RA on other purified Trichoplax NRs could be carried out to prove that TaRXR is the only nM-affinity receptor for 9-cis-RA. Another possibility might be that the Trichoplax NRs could form heterodimers.

6) In the section about expression pattern of NRs (reported in the text and in Fig. 5) in Trichoplax, the authors write (line 346) "From the experience we gained by culturing T. adhaerens, as well as from the previous experiments we knew that the culture conditions can dramatically influence phenotype. Having the possible developmental functions of the ancestral NRs in mind, we raised the question whether the expression patterns of the NRs reflect changes in phenotype." To me it is not entirely clear what this statement is supposed to mean. Why wasn't the expression level of different NRs established first on a normal, large enough population of animals with different sizes (i.e. before selecting animals with different sizes)? Is the difference in the expression profile in small and big (large) animals really significant (33% of RXR compared to other NRs in big and 24% in small animals)? And most importantly: what was counted as big and what as small animal? How was the size of the animals determined? Which animal population, the big or small or both sizes, showed an increase to 51% after treatment (line 354)? In the Methods it is written that the animals were treated only during sunny days. Is this an important information as all 9-cis-RA binding experiments have been performed in the dark? I explicitly ask this, because in the following part of the manuscript, it is reported that the feeding conditions have a visible impact on the animals. This statement is not supported by a rigorous analysis, however. Some pictures of animals in different shapes and colours are shown in Fig. 6, which, in my opinion, solely document that the animals can vary, but it remains unclear whether the variation was induced by a certain feeding condition. It also remains unclear whether the poor growth of Trichoplax on Chlorella (Fig. 7), for example, is due to the low nutritive value or composition of that particular algae or simply because this algae is not the preferred food source of Trichoplax or because the algae is better protected against being captured by the animal. In that case, Trichoplax would just eat less. In Fig. 8, I did not understand what exactly was plotted. In some panels it might be the number of animals and in others the area, as there is no legend given. Have the growth experiments been carried out only once for each condition, because there is again no error bar given (Fig. 7 and Fig. 8)?

Experimental design

no comment

Validity of the findings

no comment

Additional comments

no comment

·

Basic reporting

no comment

Experimental design

no comment

Validity of the findings

no comment

Additional comments

Novotney et al studied an ancestral Trichoplex RXR ortholog and find that it is activated by 9-cis-RA leading to the expression of other nuclear receptors. This is a neat report. To my mind it is the first example of a nuclear receptor ortholog in a basal metazoan being activated by RA.

Comments
1. The effect of algal extract on gene expression is very interesting. Phytochemicals are known to activate the estrogen receptor. Thus, I have a bias for phytochemicals also activating RXR. However, phytochemicals could activate pathways that increase the synthesis of retinoids or other novel endogenous molecules that are ligands for RXR. This is a strong alternative possibility. Considering the strong sequence similarity between Trichoplax RXR and human RXR, the extract should affect human RXR action, an experiment that would provide support for Novetney et al.

2. RA binding to RXR regulating other nuclear receptors in Trichoplax is very interesting. Activation occurs at 1 nM with a clear decrease at higher concentrations. Two thoughts: first, assays at 0.5 nM and 0.1 nM would be of much interest. Regulation at these concentrations would increase the impact of this research. Second, a comment on the decreased activity at relatively low concentrations of RA [3nM] is appropriate. It suggests to me that RA is not the endogenous ligand and thus, Trichoplax does not have metabolic pathways to regulate RA homeostasis as is found for steroid homeostasis in vertebrates.

3. I think the Introduction could be better focused on early metazoans, such as Trichoplax, Jellyfish and sponges. The discussion of invertebrates such as C. elegans and Drosophila could be condensed and moved to the final Discussion or eliminated. The strength of this paper is its data on the simplest animals. Trichoplax contains 4 nuclear receptors, in contrast to C. elegans and Drosophila which are more complex and not as ancient.

4. In 2008, I published a paper in BBRC [Trichoplax, the simplest known animal, contains an estrogen-related receptor but no estrogen receptor: Implications for estrogen receptor evolution 375, 623-627 (2008)] analyzing ERR in Trichoplax. As part of my analysis I did BLAST searches that showed that Trichoplax contained orthologs of RXR, HNF4 and COUP. Like Novotney et al, my BLAST searches found that Trichoplax RXR had very strong [unexpected] sequence similarity to human RXR.

5. The strong conservation of Trichoplax RXR ortholog and mammalian RXR deserves some comment. When I was studying ERR in Trichoplax, I was surprised at the strong similarity between Trichoplax RXR and human RXR. The BLAST scores were off the charts. In vertebrates, RXR functions as a heterodimer with several nuclear receptors. The requirement to bind several diverse nuclear receptors would be expected to constrain the RXR sequence. RXR forms heterodimers with Ecdysone receptor. You also cite evidence that jellyfish RXR forms heterodimers with TR. This indicates that heterodimers of RXR evolved early in metazoan evolution, which could explain the constraints on sequence and the strong similarity of Trichoplax RXR and human RXR. At least that is one simple explanation.
Moreover, I think there should be a comment about RXR acting as a homodimer instead of a heterodimer, with a transition to heterodimer in the evolution of more complex metazoans. This also implies that functioning of RXR in heterodimers of nuclear receptors was an important event in constraining sequence changes in RXR. Sometime in the future it would be worth determining if Trichoplax RXR forms heterodimers with human TR.

6. Line 95 mentions that an ancestral NR possessing gene regulatory capacity, may have been an unliganded molecular regulator. This is, to me, a reasonable possibility. Do the authors have a reference to this hypothesis?

7. Line 99. Is this a jellyfish? I would like a little more description of the gene and what was found.

8. Line 101 replace homology with identity.

9. Line 38, 125 and numerous other places. I think that orthologs are more accurate than homologs. To me, two genes are orthologs if they are descended from the same gene by speciation. Usually they have the same function, although I think it likely that that some orthologs may acquire a new function over time. As I understand it, the main point of this manuscript is that Trichoplax has a gene that has the same ligand binding activity at human RXR, as well as strong sequence identity. Thus, Novotny et al are proposing that the Trichoplax gene is an ortholog of vertebrate RXR. Also consider replacing homolog with ortholog in other places in the manuscript.

·

Basic reporting

Good.

The title should be changed as the data does not show an "endocrine-like network". It shows potential components of a potential network but there is no proof of connectivity and no proof of endocrine-likle function.

Experimental design

Good

Validity of the findings

Good

Additional comments

Review of “Trichoplax adhaerens reveals an endocrine-like network sensitive to 9-cis-retinoic acid at the base of metazoan evolution (#6219)”

Thanks to the authors for submitting this paper. The English is excellent and concise, the figures and headings are clear, and the subject interesting. I suggest the paper in accepted for publication with some minor changes. I have attached a copy of the PDF with changes in notes on the document but these are also detailed below. My main criticism of the manuscript is that parts of the discussion (and some of the results section) try to push potential ideas and suggestions of function a little too far. As such some of the conclusions should be softened and some of the rhetoric about potential function should be removed.

In addition the title must be changed to reflect the comments made in the conclusions (the data does not show an "endocrine-like network". It shows potential components of a potential network but there is no proof of connectivity and no proof of endocrine-likle function.)

See pdf for accurate corrections…
L 104 - NR3 should be NR3B (There are probably better refs but this is clear in Vogeler 2014; DOI: 10.1186/1471-2164-15-369)
L137 - correction (using becomes implemented in)
L139 – please clarify number of animals in each extraction
L168 – I think g. Not xg.
L209 – GPS coordinates have too many decimal places.
L283 – Delete “a high”
L294 - These two lists are the same.
L308 - With Nuclear receptors I do not find these whole gene identities particularly informative. I would much prefer to see Identity figures for each of the LBD and DBD alongside the whole gene identity, as these give much better idea of functional variability between species.
L383 – replace “more affecting” with “influencing”
L384 – insert more “… more than”
L400 - insert known “arrangement known with”
L402 – Dispose? Do you mean display? Or perhaps "...are disposed with..."
Comments
L423 - I agree but I suppose it depends on the assumed phylogenetic position of the phylum Placozoa (as mentioned above).
L438-L458 The evidence you have presented is convincing. However (and you have eluded to this), it is still not proof enough and there is a substantial link missing here in cause and effect relationship.

I think you need to soften your stance, clarify exactly what is proven and perhaps suggest what needs to be done experimentally to prove the signalling cascade you are hypothesizing.


To clarify...

Proven...
RXR binds 9-cis-RA.
9-cis-RA exposure increases expression of metabolic gene.

Hypothesis (not proven)...
9-cis-RA binds RXR and induces expression of metabolic gene.

L460 - This title should be changed in line with comment below (L479) (The data only shows a connection in expression levels upon treatment. It does not demonstrate a regulatory connection or formation of a network)

L479-481 - I find this a stretch. The data only shows a connection in expression levels upon treatment. It does not demonstrate a regulatory connection.

L507-L510 - The hormone story is perhaps a bit of a stretch (a hormone is a signalling molecule produced by the animal itself; food can mimic a hormone but it isn't a hormone). A more likely hypothesis would be that certain food contains essential nutrients at levels that allow T.a. to grow more efficiently. This is the case for almost all algal feeding inverts and is well characterised in commercial bivalve species.

L511-512 This is a really nice (and probably the most important) finding.

L512-L516 This is the logical next step but the work in this paper does not go this far. I think you should soften the rhetoric in the second half of this paragraph. Or simply even delete it.

L529 - What endocrine-like network? There is no endocrine like network proven. Can you write this instead "In conclusion, the presence of functional NRs in T. adhaerens supports ..."

---

## Round 0.2 · accepted · Accept

Many congratulations! I look forward to seeing the formatted finished article in press. Kind regards, Chris

·

Basic reporting

no comment

Experimental design

no comment

Validity of the findings

no comment

Additional comments

This is a much improved manuscript, starting with the new title. They have made numerous edits in response to the other reviewers, and this improves the manuscript. Although the authors decided not to do additional experiments that I suggested, the data and conclusions are sufficient to merit publication; that is, I think this is a useful contribution to the literature. It is a long manuscript covering both molecular studies and feeding studies. This may influence the readership.

I hope there will be follow-up studies of experiments suggested in my original review.